# Development of Halotolerant Microbial Consortia for Salt Stress Mitigation and Sustainable Tomato Production in Sodic Soils: An Enzyme Mechanism Approach

Thukkaram Damodaran [1,*], Sunil Kumar Jha [1], Sangeeta Kumari [1], Garima Gupta [1,2], Vinay K. Mishra [1], Parbodh C. Sharma [3], Ram Gopal [1], Arjun Singh [1] and Hanuman S. Jat [3]

[1]   ICAR Central Soil Salinity Research Institute, Regional Research Station, Lucknow 226002, India
[2]   Institute of Biosciences and Technology, Shri Ramswaroop Memorial University, Lucknow 225003, India
[3]   ICAR Central Soil Salinity Research Institute, Karnal 132001, India
[*]   Correspondence: t.damodaran@icar.gov.in

**Abstract:** Salt stress caused by sodic soils is an important constraint that impacts the production of crucial solanaceous vegetable crops globally. Halotolerant poly-extremophiles rhizobacteria can inhabit hostile environments like salinity, drought, etc. The present study was aimed to design a halotolerant micro-formulation using highly salt-tolerant bacterial strains previously isolated from salt-tolerant rice and wheat rhizosphere in sodic soil. Nine halotolerant isolates were examined for plant growth-promoting traits and biomass production in pot studies with sodic soil of pH 9.23 in tomato. Compatible, efficient isolates were aimed to be formulated into different consortia like PGPR-C1, PGPR-C2 and, PGPR-C3 for field evaluation in sodic soils of pH 9.14. Halotolerant rhizobacterial consortia (PGPR-C3) comprising *Lysinibacillus* spp. and *Bacillus* spp. were found to produce extracellular enzymes like amylase, protease, cellulase, and lipase, showing significantly enhanced vegetative parameters, yield and lycopene content of tomato hybrid NS585 under salt-stressed sodic soils. PGPR-C3 consortia also showed enhanced plant growth-promoting activities and halo tolerance like high Indole acetic acid production, 1-aminocyclopropane-1-carboxylic acid deaminase, and antioxidative enzyme activity over the uninoculated control. Further, inoculation with PGPR-C3 consortia resulted in the efficient exclusion of $Na^+$ ions from the rhizosphere through increased absorption of K+. Results of the study reveal that inoculation with PGPR-C3 consortia could alleviate the salt stress and promotes the successful cultivation of tomato crop in sodic soils. It can be considered the best option for eco-friendly, sustainable cultivation of vegetables like a tomato in sodic soils with a high pH range of up to 9.14.

**Keywords:** 1-aminocyclopropane-1-carboxylic acid deaminase; halotolerant PGPR consortia; indole acetic acid production; tomato-soil sodicity; salt stress; soil enzyme

## 1. Introduction

Salinity is one of the most devastating abiotic stresses for sustainable agricultural practices that show deleterious effects on soil processes, plant physiology, growth, productivity, and quality of crops [1]. Effects of climate change, like extended dry periods and high temperatures, lead to an increase in the evapotranspiration rate, resulting in soil salinity [2]. High salinity reduces soil biodiversity, microbial activity [3], and the availability of essential nutrients and minerals to plants [4,5], resulting in decreased seed germination, photosynthetic potential, and plant growth [6,7]. Sodic soils have a pH greater than 8.5, with an exchangeable sodium percent greater than 15 and are dominated by free carbonates and bicarbonates with poor hydraulic conductivity, impeding crop root growth [8]. About 60% of salt-affected soil in India is sodic or saline-sodic [9]. Variation in soil properties due to sodicity affects the microbial population in rhizospheric soil, hindering the growth of salt-sensitive commercial crops [10]. The tomato is an important vegetable crop known

for its nutritional value and food security. However, it is susceptible to salt stress, making cultivation in sodic soils difficult. The canal command areas of the Indo-Gangetic plains that were earlier known for their cultivation of important crops like gladiolus and tomato were constrained by the development of sodicity, which hindered their cultivation [11].

Earlier, various physical and chemical methods, such as scraping, flushing, and leaching, were used to remove soluble salts from the root zone [12]. The use of gypsum reclaims the sodic soil. Still, it does not contribute to the increase in the rhizospheric microbial population [13], suggesting the supplementation of microbe-mediated interventions to develop a favorable rhizosphere for the successful cultivation of crops [14]. Halotolerant plant growth-promoting rhizobacteria (H-PGPR) have been considered a viable option for sustainable production in sodic soils [15]. Studies on H-PGPR confirmed that they release a wide range of phytohormones, including gibberellins [16], cytokinins, and auxins [17–19], as well as secondary metabolites and osmolytes like exopolysaccharides [20,21], trehalose, proline, and glycine betaines [22,23] that activate plants' antioxidative enzymes (Peroxidase (POD),Superoxide dismutase (SOD), Catalase (CAT), NR (Nitrate reductase) and GR (Glutathione reductase)) under salt stress conditions [24,25]. Microbial consortia perform various crucial tasks, such as enhancing plant growth by functioning as an osmoprotectant and antioxidant to reduce salt stress. The consortium of salt-tolerant PGPR enhanced the auxin synthesis and regulated the absorption of $Na^+$, $K^+$, and $Ca^{2+}$ [26]. Research shows that microbial consortium administration is more advantageous for improving plant stress than applying them alone [27,28].

Further research is required to substantiate and confirm that microbial consortia are superior to individual isolates. The study's primary goal was to develop halotolerant microbial consortia to ameliorate sodicity stress in tomatoes. In the current research, we analyzed nine halotolerant rhizobacterial isolates that had been isolated and characterized for salt tolerance. The present study involved successive laboratory, pot, and field experiments using the potential salt tolerant isolates, based on the hypothesis that bacterial inoculation influences plant growth by stimulating antioxidant machinery under salt stress conditions. Therefore, combined inoculation of bacterial inoculants would be more effective than individual strains in regulating and activating defense mechanisms related to salt tolerance in tomato.

## 2. Materials and Methods

### 2.1. Isolates Used in the Study

Nine isolates i.e., *Pseudomonas* sp. CSR-C1 (KU867577), *Agromyces tropicus* CSR-C5 (KU860548), *Bacillus flexus* CSR-C7 (KU855379), *Bacillus tequilensis* CSR-C9 (KU860545), *Pseudomonas mendocina* CSR-C10 (KU860546), *Lysinibacillus sphaericus* CSR-A16 (KU745625), *Bacillus subtilis* CSR-A18 (B.I-768236), *Lysinibacillus fusiformis* CSR-A11 (KU745624), and *Bacillus licheniformis* CSR-M16 (KC768636) were obtained from ICAR CSSRI, RRS, Lucknow, India. The strains were previously isolated from rhizospheric soil of salt-tolerant wheat (KRL-210) and rice (CSR-36) cultivars raised in sodic soils (pH range from 9.40–10.20) at the Hardoi, Lalganj, Sitapur, Kanpur Dehat, and Vinoba Nagar districts in Uttar Pradesh, India. The isolates used were earlier identified using 16S rDNA sequencing [29].

### 2.2. Extracellular Enzyme Assay

In the present study, the nine bacterial isolates used were screened for various enzymatic activities, including cellulase, protease, lipase, and amylase. The Gram's iodine technique was used to test the cellulose activity as prescribed by Kasana et al. [30]. Point inoculations of bacterial cultures were made onto carboxymethycellulose agar media (2 g $NaNO_3$, 1 g K2HPO$_4$, 0.5 g MgSO$_4$, 0.5 g KCl, 0.2 g peptone, 2 g CMC sodium salt, and 17 g agar in 1000 mL distilled water) and incubated at 28 ± 2 °C for four days. The appearance of clear zones around bacterial colonies after flooding the culture plates with Gram's iodine (2.0 g potassium iodide and 1.0 g iodine in 300 mL distilled water) was considered to be positive for cellulase activity. Similarly, proteolytic activity was analyzed as per Kavitha

et al. [31] with slight modification. Bacterial isolates were spotted onto a skim milk agar plate (10 g casein hydrolysate, 5 g yeast extract, 4 g NaCl, 20 g skim milk powder, and 10 g agar in 1000 mL distilled water) and incubated at 28 °C for 5 days. The formation of a clear halo zone around the colony shows positive protease production. Similarly, bacterial cultures were spot inoculated on spirit blue agar with 3% Bactolipase reagent and incubated for 48 h at 35 °C for the lipase test. The formation of the zone around the colony shows positive lipase activity [32]. The amylase enzyme activity was measured in NA media containing starch (1%). About 24 h old cultures of bacterial isolates were streaked on starch-containing NA plates and incubated for 48 h at 35 °C. The appearance of a clear zone after adding Gram's iodine solution to the starch-containing medium is considered positive amylase activity [32].

### 2.3. Effect of PGPR Isolates on Tomato Hybrid NS585 in Sodic Soil

The nine H-PGPR bacterial isolates were evaluated for sodicity tolerance in tomato *var.* NS585 in pots with soil pH 9.23 and EC 0.598 for growth promotion and shoot root biomass. The soil used in the pot study was sterilized in an autoclave before the trial. Each bacterial isolate was grown separately on a rotary shaker (Vaiometra, Associated scientific technologies) for 48 h at 150 rpm at $28 \pm 2$ °C in a 250 mL conical flask containing CSR culture media (No. 3857/DEL/2012) [33]. The purity of the isolates was checked by the streaking process. After 48 h of incubation, bacterial cells were collected via centrifugation (Micro 17 TR, Micro high-speed centrifuge) at 10,000 rpm for 15 min at 4 °C and adjusted to $6 \times 10^8$ CFU mL$^{-1}$ using sterilized double distilled water. The tomato seeds were treated with a 1% prepared bacterial suspension for about 2 h before transplantation and, further, for foliar application on the 20th day after sowing. The seedlings treated with sterilized distilled water were considered the control. The trial was laid out with five replications, following a complete randomized block design. After 40 days of stress, the tomato plants were observed for different growth parameters like plant height, root-shoot fresh weight, and dry weight. The samples (root and shoot) were dried in a hot air oven at 65 °C for about 96 h to determine the dry weight.

Simple ranking and scoring systems based on plant height and root-shoot fresh and dry weight were used to select the four effective H-PGPR isolates, formulated as bio-consortia [34], for further evaluation in field experiments in sodic soils. The simple scoring and ranking guidelines are as follows: (i) the shortest plant height received a score of 1, the higher received a score of 2, and so on; (ii) the least root-shoot fresh weight was given a score of 1, the higher was given a score of 2, and so on; (iii) the least root-shoot dry weight received a score of 1, the higher received a score of 2, and so on; (iv) all scores were added, and ranking was done based on the highest of the total score; (v) the highest overall score was rank 1, followed by rank 2, and so on.

### 2.4. Bio-Formulation of Microbial Consortia

Highly efficient halotolerant rhizobacterial isolates were selected based on their enzymatic assay and plant growth potential in sodic soil. The formulation of effective microbial consortia as inoculants requires evidence of the ability of the consortium members to be compatible; therefore, microbial isolates were subjected to an in vitro compatibility test using the streaking technique. Each isolate was streaked vertically and horizontally on nutrient agar media. Plates were incubated at $28 \pm 2$ °C and observed at 24 h intervals over a period of 4 days. They were identified as incompatible in case there is formation of lysis or inhibition zone, whereas when no inhibition zone was observed, the isolates were identified as compatible. The compatible H-PGPR were formulated into consortia and mass multiplied as per the protocol followed by Damodaran et al. [35] for further studies.

### 2.5. Screening of Salt Tolerance and Sodium Uptake

All isolates were cultured in test tubes for 24 h using an incubated rotary shaker (120 rpm) at $28 \pm 2$ °C. After that, compatible individual cultures ($10^7$ CFU/mL) were

mixed in uniform proportions to form consortia. The intrinsic sodicity tolerance ability of bacterial consortia was assessed by inoculating the 250 μL bacterial consortia into the nutrient broth (NB) (10 mL) supplemented with 5.0%, 7.5%, and 10.0% NaCl and incubated for 24 h at $28 \pm 2\ °C$ at 120 rpm. Bacterial growth was determined through a spectrophotometer (UV-Vis Spectrophotometer, UV-1900i) at 600 nm ($OD_{600}$), and uninoculated liquid media was used as a blank to find out the NaCl tolerance level. Sodium uptake pattern was assessed among the cultures showing luxurious growth in NaCl-rich media. To analyze the sodium uptake pattern, the bacterial cells were collected after 24 h by centrifugation and washed with sterilized distilled water. The washed pellet was digested overnight with 0.1 N HCl, and the supernatant was taken for the determination of sodium uptake by bacterial cells using a flame photometer (Flame Photometer 128, Systronics) [34].

### 2.6. Quantification of ACC Deaminase Activity

The ACC deaminase activity was quantitatively assessed at 540 nm using a spectrophotometer (UV-Vis Spectrophotometer, UV-1900i) in terms of α—ketobutyrate production and compared to the standard curve (0.1 to 1.0 μmol) of α—ketobutyrate [36]. The Bradford method was used for protein estimation [37]. One unit of ACC deaminase activity was expressed as the amount of α—ketobutyrate liberated in nmol per milligram of cellular protein per hour.

### 2.7. Indole Acetic Acid Production (IAA)

The rhizobacteria were subcultured in NB media amended with tryptophan (5 mM) and incubated at 28 °C for 7 days at 200 rpm. Salkowski reagent (0.5 M $FeCl_3$ + 70% perchloric acid) was used to assess IAA production using the colorimetric method. The addition of Salkowski reagent and culture supernatant (4:1) develops the red color due to Indolic compound synthesis, which was measured by UV-vis spectrophotometer at 530 nm [38]. A pure IAA standard curve with a range of 0 and 100 μg mL$^{-1}$ was used to calculate the IAA concentration.

### 2.8. Effect of Different Microbial Consortia on Tomato Hybrid NS585

In the tomato *var.* NS585, the formulated consortia were evaluated in terms of growth and yield variables. Tomato seedlings were transplanted after 35–40 days of showing in the main field of sodic soil (pH 9.14) (Table 1) at the experimental research farm of ICAR-CSSRI, RRS, Lucknow, India.

**Table 1.** Initial physical and chemical characteristics of experimental field soil.

| Soil Parameters | Values |
| --- | --- |
| Soil pH | 9.14 |
| EC | 0.52 ds m$^{-1}$ |
| Sodium ($Na^+$) | 15.79 meq L$^{-1}$ |
| Potassium ($K^+$) | 0.13 meq L$^{-1}$ |
| Carbonate | 2.63 meq L$^{-1}$ |
| Bicarbonate | 1.62 meq L$^{-1}$ |

Note: EC—Electrical conductivity, $Na^+$—Sodium cation, $K^+$—Potassium cation of soil.

The experiment used a randomized block approach with a 5 m × 5 m plot size, grown during two consecutive rabi seasons (November–February) of the year 2015–16 and 2016–17. The bacterial cultures were prepared as per the methodology described in Section 2.3. To monitor the effect of different bacterial consortium on plant growth and yield, biometrical traits like plant height (cm), root fresh weight (g), root dry weight (g), shoot fresh weight (g), and shoot dry weight (g) were recorded in triplicates. Upon maturity of the crop, fruit yield/plant (kg), fruit yield/ha (tonne) was determined. The lycopene content in the fruit under different treatments was also analyzed using the rapid hexane-extraction method in a spectrophotometer (UV-Vis Spectrophotometer, UV-1900i) [39].

### 2.9. Estimation of Na/K ion Uptake by Plant

Plants from each treatment were collected in triplicates and divided into root and leaf parts. Then, the roots were blotted using soft tissue paper for drying. After that, root tissues were kept in a microfuge tube at $-70$ °C. After thawing, a hole was drilled with a needle in the lower portion of the tube and then kept in an empty microfuge tube for centrifugation for 5 min. After centrifugation, fluid was accumulated in the bottom tube. Further, the fluid was diluted as per the requirement for estimation of $[K^+]$ and $[Na^+]$ using a flame photometer [40].

### 2.10. Quantification of Plant Stress Enzymes and Compounds against Salinity Stress

For estimating the activities of stress-related enzymes, about 5 g of leaves were collected from 40-day-old tomato plants for each enzymatic test and kept in cool boxes for transportation. The proline was estimated using the sulphosalicylic acid extraction method using the reaction mixture of acid ninhydrin (2 mL) and glacial acetic acid (2 mL) [41]. The total soluble sugar content in the plant leaves was analyzed using the anthrone protocols, where leaf samples were digested in hydrochloric acid and neutralized with sodium carbonate, as described by Thimmaiah [42]. Superoxide dismutase (SOD) activity (EC 1.15.1.1) was determined as described by El-Moshaty et al. [43] and was expressed in units/g tissue. SOD activity was assessed by measuring the supernatant capacity to prevent the photochemical degradation of nitroblue tetrazolium (NBT). The test sample (3 mL) contains 75 μM NBT, 2 μM riboflavin, and 13 mM methionine in a 50 mM/L sodium phosphate buffer with a pH of 7.8. Then, 100 μL of the enzyme extract and riboflavin were added, along with 0.1 mM EDTA. The peroxidase (POD) activity analysis was done using the steps outlined by Hammerschmidt et al. [44]. The reaction mixture contained 10 mM hydrogen peroxide, 0.25% v/v guaiacol in a 10 mM potassium phosphate buffer of pH 6.9, and 0.1 mL of enzyme extract. The POD was determined by measuring absorbance at 470 nm per min per mg protein. Polyphenol oxidase (PPO) was assayed using the modified method of Oktay et al. [45] using a reaction mixture consisting of 0.01 catechol (1 mL) in a phosphate buffer (0.9 mL) of pH 7.5 and 0.1 mL of enzyme extract. The absorbance was measured at 420 nm immediately after adding the enzyme extract.

### 2.11. Statistical Analysis

The experiment was conducted in RBD, and analysis of variance (ANOVA) was used to statistically analyze the obtained data, and Duncan's multiple range test ($p \leq 0.05$) was used to compare means using SPSS [46]. The pot experiment was conducted in five replicates, and the field experiment was conducted in three replicates.

## 3. Result

### 3.1. Extracellular Enzyme Assay

The highest enzymatic activity was found in CSR-A16 followed by CSR-A11, CSR-M16, and CSR-A18, respectively. Another isolate, CSR-A11, showed high amounts of protease, cellulase, and lipase production, and less amount of amylase. Similarly, isolate CSR-M16 showed high production of amylase, protease, and cellulase but low lipase production. The isolate CSR-A18 showed high cellulase production and low amylase, protease, and lipase production. Three isolates CSR-C1, CSR-C5, and CSR-C10 did not show any enzymatic activity (Table 2).

**Table 2.** Extracellular enzyme activity in salt tolerant bacterial isolates.

| Isolates Name | Isolates Code | Amylase | Protease | Cellulase | Lipase |
|---|---|---|---|---|---|
| *Pseudomonas* sp. | CSR-C1 | - | - | - | - |
| *Agromyces tropicus* | CSR-C5 | - | - | - | - |
| *Bacillus flexus* | CSR-C7 | ++ | - | - | + |
| *Bacillus tequilensis* | CSR-C9 | ++ | - | ++ | - |
| *Pseudomonas mendocina* | CSR-C10 | - | - | - | - |
| *Lysinibacillus sphaericus* | CSR-A16 | ++ | ++ | ++ | ++ |
| *Bacillus subtilis* | CSR-A18 | + | + | ++ | + |
| *Lysinibacillus fusiformis* | CSR-A11 | + | ++ | ++ | ++ |
| *Bacillus licheniformis* | CSR-M16 | ++ | ++ | ++ | + |

"-" means no production, "+" means low production, "++" means high production.

### 3.2. Effect of H-PGPR Isolates on Tomato Hybrid NS585 in Sodic Soil

All H-PGPR isolates could promote plant height significantly ($p \leq 0.05$). The inoculation of the H-PGPR isolates in tomato plants significantly enhanced the growth, shoot and root biomass production compared to the control (Table 3). The isolates CSR-M16, CSR-A16, CSR-A18, CSR-A11, and CSR-C10 showed 72% (71.85 cm), 69% (66.37 cm), 69% (66.33 cm), 69% (65.93 cm) and 65% (57.57 cm) improvement in plant height. The minimum plant height was recorded in the control (20 cm). Similarly, shoot and root fresh weight increased significantly with the H-PGPR inoculation. The maximum shoot fresh weight was observed in plants treated with the isolates CSR-A11 (68%), followed by CSR-M16 (68%), CSR-A16 (67%), and CSR-A18 (66%), than the control. Tomato plants treated with isolate CSR-M16 exhibited a significant ($p \leq 0.05$) increase (85%) in root fresh weight followed by CSR-C10 (85%), CSR-A11 (84%), and CSR-A16 (84%), as compared to the uninoculated control plants (Table 3). The shoot and root dry weight showed a significant increase upon inoculation with H-PGPR isolates. The isolates CSR-M16 (9.02 g and 7.73 g) and CSR-A11 (9.02 g and 6.82 g) represented a higher impact than others. Significant lower shoot dry weight and root dry weight (3.72 g and 1.49 g) were observed in the uninoculated control plants than in the plants inoculated with H-PGPR isolates, just like the other parameters (Table 3).

**Table 3.** Plant growth parameters of tomato plants *var.* NS585 treated with H-PGPR isolates.

| Treat. | Plant Parameters | | | | |
|---|---|---|---|---|---|
| | Pl. Ht. (cm) | SFWt. (g) | SDWt. (g) | RFWt. (g) | RDWt. (g) |
| CSR-C1 | 32.23 b | 45.38 cd | 5.40 bc | 8.90 b | 1.66 a |
| CSR-C5 | 38.37 cd | 43.78 c | 4.45 ab | 10.33 b | 2.03 ab |
| CSR-C7 | 39.97 d | 47.86 d | 5.56 bc | 3.97 a | 2.83 b |
| CSR-C9 | 36.33 c | 40.71 b | 3.57 a | 9.67 b | 1.93 ab |
| CSR-C10 | 57.57 e | 63.24 e | 6.54 cd | 17.57 de | 6.40 d |
| CSR-A16 | 66.37 f | 75.73 fg | 7.54 de | 16.43 d | 6.43 d |
| CSR-A18 | 66.33 f | 73.02 f | 8.07 ef | 14.57 c | 4.83 c |
| CSR-A11 | 65.93 f | 78.37 g | 9.02 f | 17.23 de | 6.82 de |
| CSR-M16 | 71.85 g | 78.28 g | 9.02 f | 18.57 e | 7.73 e |
| Control | 20.34 a | 24.70 a | 3.72 a | 2.70 a | 1.49 a |
| F value | 488.88 * | 344.31 * | 23.02 * | 135.49 * | 50.23 * |

Notes: Values are the means of three replicates. Significant differences between treatments are shown by means denoted by different letters ($p < 0.05$) (DMRT). F values marked with asterisks (*) are significant at alpha $p \leq 0.05$. Pl.Ht.- plant height, SFWt.- shoot fresh weight, SDWt.- shoot dry weight, RFWt.- root fresh weight, RDWt.- root dry weight.

Based on ranking methods including plant height, shoot and root fresh and dry weights, isolates CSR-M16, CSR-A11, CSR-A16, and CSR-A18 obtained the highest score, sequentially ranking 1, 2, 3, and 4 (Figure 1).

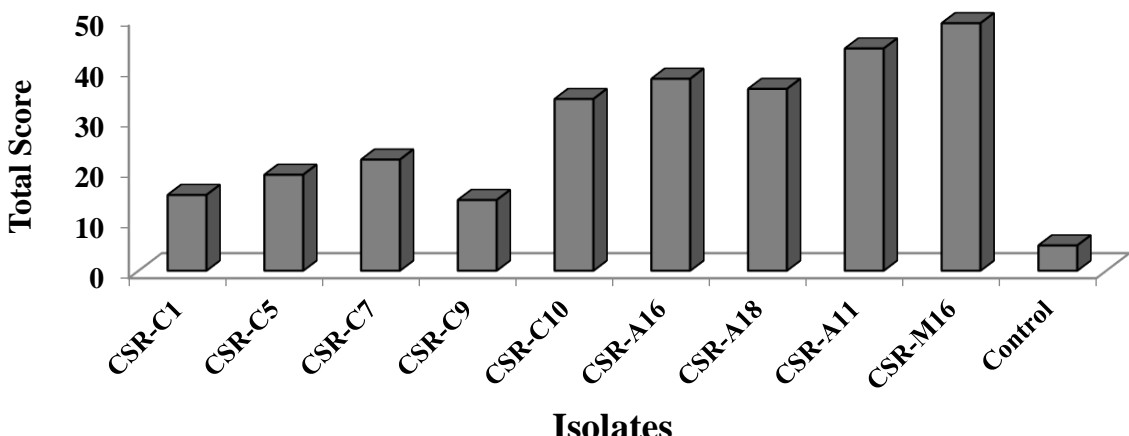

**Figure 1.** Graph showing total score of H-PGPR isolates based on plant height and the fresh and dry weights of root and shoot.

### 3.3. Compatibility Assessment and Formulation of Bio-Consortia

The compatibility test showed that CSR-A11, CSR-A18, CSR-M16, and CSR-A16 were compatible. Further, based on the extracellular enzyme activities, vegetative growth parameters in sodic soils, and the compatibility tests, three consortia combinations were formulated, entitled PGPR-C1 (*L. fusiformis* (CSR-A11) + *B. subtilis* (CSR-A18) + *B. licheniformis* (CSR-M16)), PGPR-C2 (*L. fusiformis* (CSR-A11) + *L. sphaericus* (CSR-A16) + *B. subtilis* (CSR-A18)), and PGPR-C3 (*L. fusiformis* (CSR-A11) + *L. sphaericus* (CSR-A16) + *B. licheniformis* (CSR-M16)). Finally, the bacterial consortia were mass multiplied according to the protocol described using CSR media (No. 3857/DEL/2012) for inclusion in field trials [29].

### 3.4. Screening of Salt Tolerance and Sodium Uptake

Among all consortia, the PGPR-C3 (1.650) was found to be highly salt tolerant up to NaCl (10.0%) concentration, followed by PGPR-C1 (1.504) and PGPR-C2 (1.377) respectively (Figure 2a). Similarly, at 5.0% and 7.5% NaCl concentrations, PGPR-C3 showed the highest salt tolerance followed by PGPR-C1 and PGPR-C2. All the isolates showed the higher sodium uptake PGPR-C3 (39.28%), PGPR-C1 (38.87%), and PGPR-C2 (37.13%) at 5.0% NaCl concentration (Figure 2b). At 7.5% and 10.0% NaCl concentrations, the highest sodium uptake was observed in PGPR-C3 (35.58% and 30.65%).

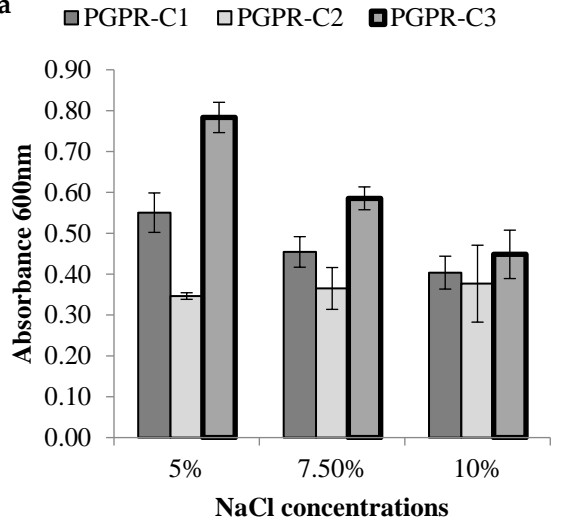

**Figure 2.** *Cont.*

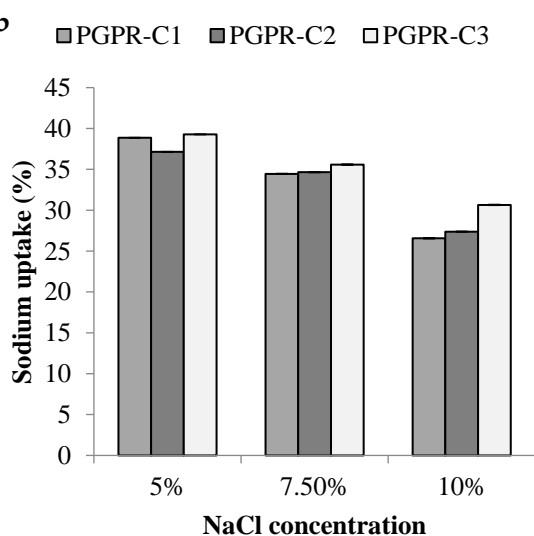

**b**

**Figure 2.** Figure showing effects of different concentrations of NaCl (5%, 7.5%, and 10%) on growth of halotolerant PGPR consortia (**a**) salt tolerance OD at 600 nm, (**b**) sodium uptake percentage. PGPR-C1: (*L. fusiformis* (CSR-A11) + *B. subtilis* (CSR-A18) + *B. licheniformis* (CSR-M16)); PGPR-C2: (*L. fusiformis* (CSR-A11) + *L. sphaericus* (CSR-A16) + *B. subtilis* (CSR-A18)); PGPR-C3: (*L. fusiformis* (CSR-A11) + *L. sphaericus* (CSR-A16) + *B. licheniformis* (CSR-M16)). Values are mean of three replicates with ± standard deviation.

*3.5. Quantification of ACC Deaminase Activity*

Data presented in Figure 3a show that the different concentrations of NaCl significantly affected ACC deaminase activity. The ACC deaminase activity ranged between 7.10 to 13.77 µM/mg protein/h α ketobutyrate. The ACC deaminase activity was enhanced by 11% and 16% by PGPR-C3 at 5% and 7.5% NaCl concentrations compared to 0% NaCl concentration. However, with the further increase in NaCl (10%) concentration, ACC deaminase activity gets reduced by 12% compared to 7.5% NaCl concentration. Low ACC deaminase activity (7.10 µM/mg protein/h α ketobutyrate) was recorded in PGPR-C1 at 0% NaCl concentration. However, it significantly increased by 34%, 64%, and 24% in the presence of NaCl at 5%, 7.5%, and 10% concentrations, respectively.

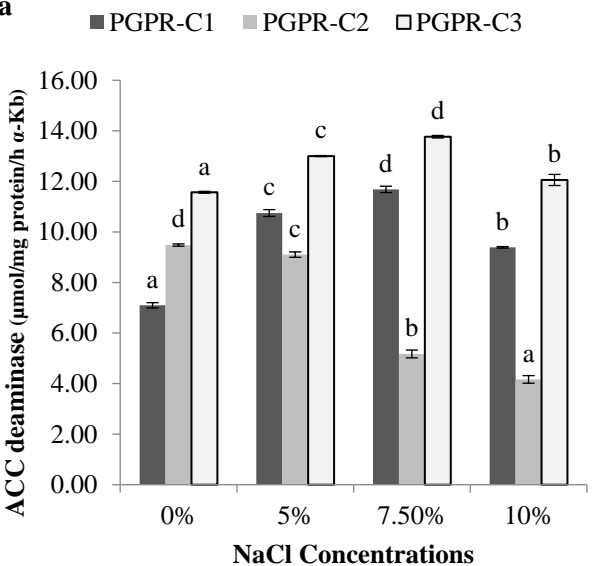

**a**

**Figure 3.** *Cont*.

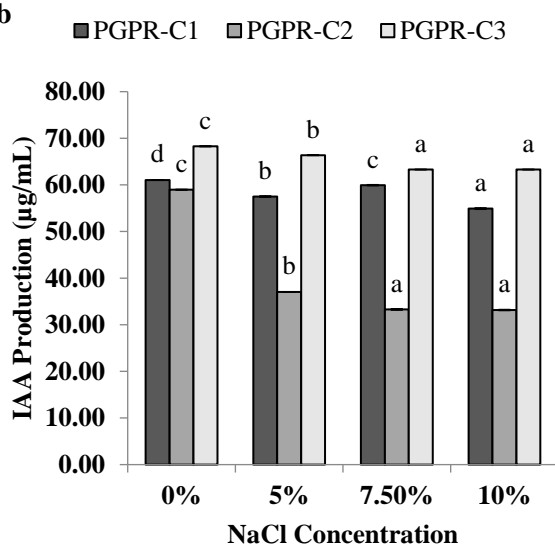

**Figure 3.** Figure showing effects of consortia of halotolerant PGPR on (**a**) 1-Aminocyclopropane-1-Carboxylate (ACC) deaminase (μmol/mg protein/h α-Kb) activity, (**b**) Indole Acetic Acid (IAA) production (μg/mL). PGPR-C1: (*L. fusiformis* (CSR-A11) + *B. subtilis* (CSR-A18) + *B. licheniformis* (CSR-M16)); PGPR-C2: (*L. fusiformis* (CSR-A11) + *L. sphaericus* (CSR-A16) + *B. subtilis* (CSR-A18)); PGPR-C3: (*L. fusiformis* (CSR-A11) + *L. sphaericus* (CSR-A16) + *B. licheniformis* (CSR-M16)). Values are mean of three replicates with ± standard deviation. Significant differences between treatments are shown by means denoted by different letters (*p* = 0.05) Duncan's multiple range test.

### 3.6. Indole Acetic Acid Production

IAA production ranged between 33.16 to 68.28 μg mL$^{-1}$ (Figure 3b). PGPR-C3 (68.28 μg mL$^{-1}$) showed the maximum IAA production at 0% NaCl concentration followed by PGPR-C1 (61.04 μg mL$^{-1}$) and PGPR-C2 (58.97 μg mL$^{-1}$) respectively. Similarly, at 5% and 7.5% NaCl concentrations, the highest IAA production was recorded in PGPR-C3 (66.37 and 63.30 μg mL$^{-1}$). Data showed that there is a significant difference in the IAA production with the increase in the NaCl concentration.

### 3.7. Field Trials to Evaluate Effect of Different Microbial Consortia on Tomato Hybrid NS585

Field trials were conducted for two years (2015–16 and 2016–17) in the sodic soils having a pH of 9.14 using three bio-consortia treatments in the tomato variety NS585.

Sodicity had a negative impact on the untreated plant growth and yield compared to the treated plants. DMRT was used to compare the mean of each treatment with the control and observed that each treatment caused a significant (*p* ≤ 0.05) increase in growth parameters. The maximum plant growth was recorded in PGPR-C3 (78.48%), followed by PGPR-C1 (76.48%) and PGPR-C2 (67.97%) compared to untreated control plants. The most diminutive plant height recorded was 22.56 cm in the control. Similarly, each treatment caused a significant (*p* ≤ 0.05) increase in shoot fresh weight and root fresh weight compared to the control, as indicated by Duncan's multiple range tests in the field experiment (Figures 4 and 5). The highest increase in shoot fresh weight (78.33%) and root fresh weight (79%) in PGPR-C3 was observed, which was significantly higher than in the control. Salinity significantly (*p* ≤ 0.05) affected the shoot dry weight and root dry weight of untreated control tomato *var.* NS585 plant (Figure 4). The highest shoot dry weight (54.96 g) and root dry weight (33.30 g) were observed in PGPR-C3 inoculated plants, followed by PGPR-C1 (46.61 g and 26.20 g) and PGPR-C2 (32.64 g and 20.99 g), respectively.

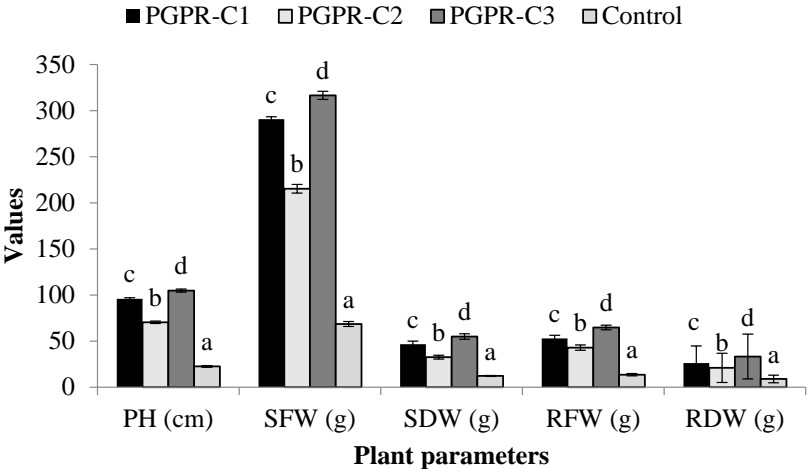

**Figure 4.** Plant growth parameters of tomato plants *var.* NS585 treated with H-PGPR microbial consortia. PH (cm)—plant height, SFW (g)—shoot fresh weight, SDW (g)—shoot dry weight, RFW (g)—root fresh weight and RDW (g)—root dry weight of tomato plants grown in the saline-stressed field of Central Soil Salinity Research Institute, Regional Research Station, Lucknow, Uttar Pradesh. PGPR-C1: (*L. fusiformis* (CSR-A11) + *B. subtilis* (CSR-A18) + *B. licheniformis* (CSR-M16)); PGPR-C2: (*L. fusiformis* (CSR-A11) + *L. sphaericus* (CSR-A16) + *B. subtilis* (CSR-A18)); PGPR-C3: (*L. fusiformis* (CSR-A11) + *L. sphaericus* (CSR-A16) + *B. licheniformis* (CSR-M16)). Mean ± SD, Significant differences between treatments are shown by means denoted by different letters (*p* < 0.05) (DMRT).

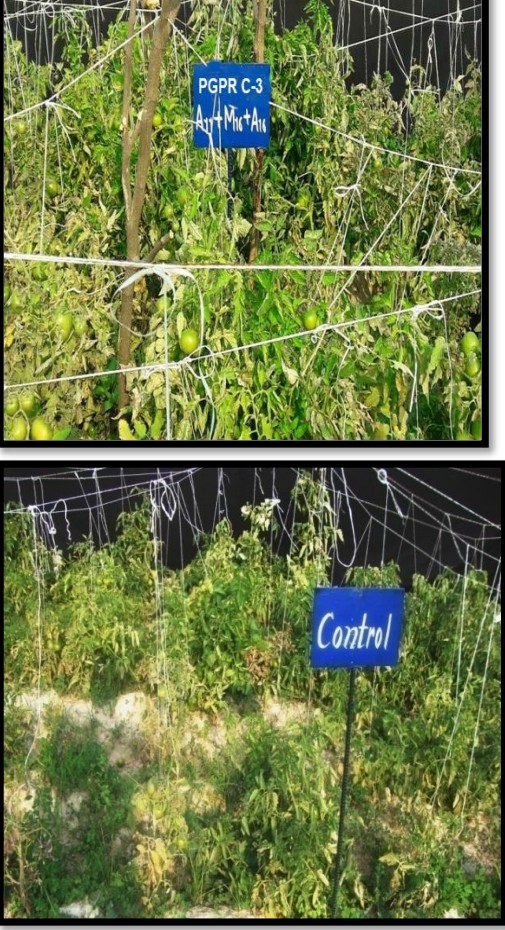

**Figure 5.** Showing field trials to evaluate the effect of PGPR-C3 microbial consortia on tomato hybrid NS585.

Percent increment in plant growth parameters of tomato plants *var.* NS585 treated with consortia, compared to H-PGPR isolates presented in Table 4, showed that the highest plant height increment in CSR-A11 (37.12%) in consortia form with CSR-A16 and CSR-M16 (PGPR-C3) compared to CSR-A11 individually, followed by CSR-A16 (36.7%) and CSR-M16 (31.47%), respectively, in consortia form PGPR-C3 compared to individual single isolate form. The minimum increment in plant height was recorded with CSR-A16 (5.76%) in consortia form with CSR-A18 (5.82%) and CSR-A11 (6.39%) (PGPR-C2) compared to in isolate form. Maximum shoot fresh weight increment was observed in CSR-A18 (74.88%) in consortia with CSR-M16 (73.07%) and CSR- A11 (73.04%) (PGPR- C1) compared to their isolate form. The highest increments in shoot dry weight was recorded with CSR-A16 (86.28%) in consortia form with CSR-A11 (83.59%) and CSR-M16 (83.59%) PGPR-C3 compared to isolate forms. Similarly, the maximum increment in shoot fresh weight and shoot dry weight was observed in CSR-M16 (76.79%) and CSR-A11 (85.5%) in consortia form with PGPR-C3, respectively.

**Table 4.** Percent increments in plant growth parameters of tomato plants *var.* NS585 treated with consortia compared to H-PGPR isolates.

| Consortia | Isolates | Percent Increment (%) | | | | |
|---|---|---|---|---|---|---|
| | | **Pl. Ht.** | **SFWt.** | **SDWt.** | **RFWt.** | **RDWt.** |
| PGPR-C1 | CSR-A11 | 31.26 | 73.04 | 80.65 | 67.48 | 81.56 |
| | CSR-A18 | 30.84 | 74.88 | 82.69 | 72.5 | 75.46 |
| | CSR-M16 | 25.09 | 73.07 | 80.65 | 64.96 | 70.5 |
| PGPR-C2 | CSR-A11 | 6.39 | 63.6 | 72.37 | 59.87 | 76.99 |
| | CSR-A18 | 5.82 | 66.08 | 75.28 | 66.07 | 69.37 |
| | CSR-A16 | 5.76 | 64.82 | 76.9 | 61.74 | 69.51 |
| PGPR-C3 | CSR-A11 | 37.12 | 25.26 | 83.59 | 73.41 | 85.5 |
| | CSR-A16 | 36.7 | 27.77 | 86.28 | 74.64 | 80.78 |
| | CSR-M16 | 31.47 | 25.34 | 83.59 | 76.79 | 76.79 |

Note: Percent increment of plant height (Pl.Ht.), shoot fresh weight (SFWt.), shoot dry weight (SDWt.), root fresh weight (RFWt.) and root dry weight (RDWt.) of tomato hybrid NS585 grown in the saline-stressed field of Central Soil Salinity Research Institute, Regional Research Station, Lucknow, Uttar Pradesh. PGPR-C1: (*L. fusiformis* (CSR-A11) + *B. subtilis* (CSR-A18) + *B. licheniformis* (CSR-M16)); PGPR-C2: (*L. fusiformis* (CSR-A11) + *L. sphaericus* (CSR-A16) + *B. subtilis* (CSR-A18)); PGPR-C3: (*L. fusiformis* (CSR-A11) + *L. sphaericus* (CSR-A16) + *B. licheniformis* (CSR-M16)).

The data of two consecutive field trials represented in Figure 6 showed that treatment with PGPR-C3 had a significant effect ($p \leq 0.05$) on total yield. In the first year, the highest yield/plant and yield/ha were recorded in PGPR-C3 (1.96 kg/plant and 5.37 ton/ha), followed by PGPR-C1 (1.56 kg/plant and 4.63 ton/ha) and PGPR-C2 (1.43 kg/plant and 3.13 tons/ha), respectively. Similarly, during the second consecutive year, PGPR-C3 showed the highest productivity. The data for two years (Figure 5) showed that the yield/plant and yield/ha were significantly higher in PGPR-C3 compared to PGPR-C1 (1.61 kg/plant and 4.82 ton/ha), PGPR-C2 (1.45 kg/plant and 3.38 ton/ha), and the control (0.29 kg/plant and 1.30 ton/ha).

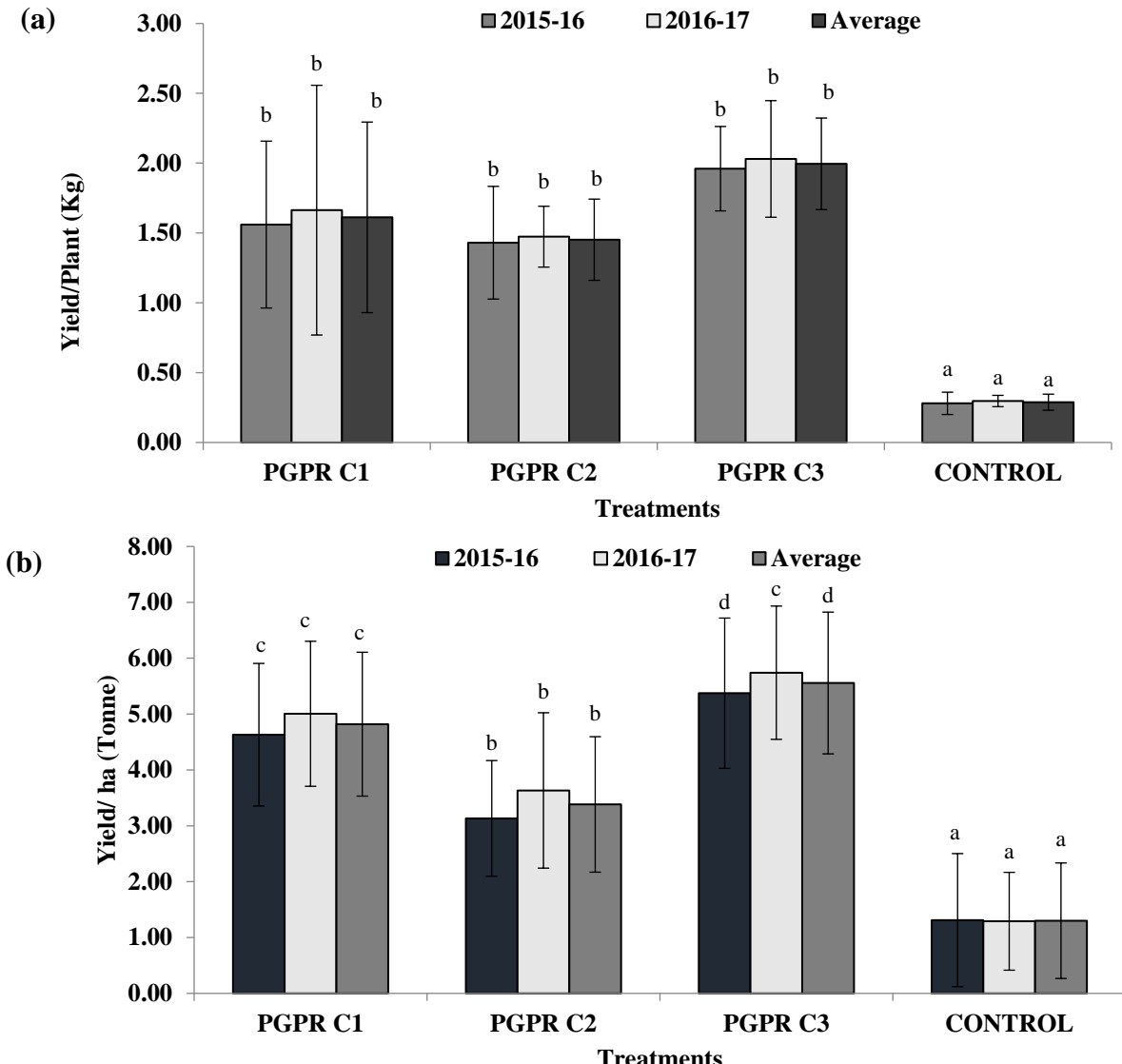

**Figure 6.** Effects of consortia of PGPR isolates on the (**a**) yield/plant (Kg) and (**b**) total yield per hectare (tonne) of tomato hybrid NS585 grown in the saline-stressed field of Central Soil Salinity Research Institute, Regional Research Station, Lucknow, Uttar Pradesh. PGPR-C1: (*L. fusiformis* (CSR-A11) + *B. subtilis* (CSR-A18) + *B. licheniformis* (CSR-M16)); PGPR-C2: (*L. fusiformis* (CSR-A11) + *L. sphaericus* (CSR-A16) + *B. subtilis* (CSR-A18)); PGPR-C3: (*L. fusiformis* (CSR-A11) + *L. sphaericus* (CSR-A16) + *B. licheniformis* (CSR-M16)). Significant differences between treatments are shown by means denoted by different letters (*p* = 0.05) Duncan's multiple range test.

The physiochemical analysis of quality parameters in tomato showed that inoculated plants exhibited maximum lycopene content (2–4 folds) compared to the control (Table 5). A significantly higher lycopene content for two consecutive years was observed in PGPR-C3 (17.23 mg/100 g and 17.19 mg/100 g), followed by PGPR-C2 (12.50 mg/100 g and 12.93 mg/100 g) and PGPR-C1 (9.52 mg/100 g and 9.67 mg/100 g), respectively. According to DMRT, the average quantity of lycopene observed in treatment PGPR-C3 (17.21 mg/100 g) was significantly ($p \leq 0.05$) higher, followed by PGPR-C2 (12.71 mg/100 g), PGPR-C1 (9.60 mg/100 g), and the control (4.03 mg/100 g).

**Table 5.** Effects of inoculation of PGPR consortia on total soluble sugar (TSS%), sodium potassium ratio (Na$^+$/K$^+$), and lycopene content in tomato var NS585.

| Year | PGPR | TSS% | Na$^+$/K$^+$ | Lycopene (mg/100 g) |
|---|---|---|---|---|
| 2015–16 | PGPR-C1 | 5.47 b | 0.21 b | 9.52 b |
| | PGPR-C2 | 5.47 b | 0.45 c | 12.50 c |
| | PGPR-C3 | 8.07 c | 0.15 a | 17.23 d |
| | Control | 3.00 a | 0.91 d | 3.98 a |
| | **Total** | **5.50** | **0.43** | **10.81** |
| 2016–17 | PGPR-C1 | 5.40 b | 0.21 b | 9.67 b |
| | PGPR-C2 | 5.43 b | 0.46 c | 12.93 c |
| | PGPR-C3 | 8.03 c | 0.17 a | 17.19 d |
| | Control | 3.07 a | 0.92 d | 4.07 a |
| Average | **Total** | **5.48** | **0.46** | **10.97** |
| | PGPR-C1 | 5.45 b | 0.22 b | 9.60 b |
| | PGPR-C2 | 5.45 b | 0.46 c | 12.71 c |
| | PGPR-C3 | 8.05 c | 0.17 a | 17.21 d |
| | Control | 3.03 a | 0.92 d | 4.03 a |
| | Total | 5.49 | 0.44 | 10.89 |

Note: PGPR-C1: (*L. fusiformis* (CSR-A11) + *B. subtilis* (CSR-A18) + *B. licheniformis* (CSR-M16)); PGPR-C2: (*L. fusiformis* (CSR-A11) + *L. sphaericus* (CSR-A16) + *B. subtilis* (CSR-A18)); PGPR-C3: (*L. fusiformis* (CSR-A11) + *L. sphaericus* (CSR-A16) + *B. licheniformis* (CSR-M16)). Values are mean of three replicates. Significant differences between treatments are shown by means denoted by different letters ($p \leq 0.05$) (DMRT).

### 3.8. Estimation of Na/K ion Uptake by Plant

The data of ionic accumulation (Na$^+$ and K$^+$) in the roots of inoculated and control plants show that salt stress considerably increased the Na$^+$ and decreased the K$^+$ uptake in the uninoculated plants compared to inoculated plants. Significantly ($p \leq 0.05$), higher amounts of Na$^+$ and lower concentrations of K$^+$ were recorded in the roots of uninoculated plants cultivated in sodic soil. The sodium ions in the root of treated tomato plants grown under salt stress conditions was significantly ($p \leq 0.05$) lower than in uninoculated plants by two to six times, respectively. The application of the PGPR consortium significantly reduces the Na$^+$ uptake and increases the uptake of K$^+$ (Table 5). The lowest Na$^+$ uptake was recorded in plants inoculated with PGPR-C3, followed by PGPR-C1 and PGPR-C2. In contrast, the maximum uptake of K$^+$ was observed in consortia PGPR-C3-inoculated plants, significantly ($p \leq 0.05$) higher than in PGPR-C1, PGPR-C2, and uninoculated control plants, respectively.

### 3.9. Quantification of Plant Stress Enzymes and Compounds against Salinity Stress

The accumulation of proline and total soluble sugar estimated in treated and untreated tomato plants grown in saline soil varied significantly ($p \leq 0.05$) between inoculated and uninoculated plants. Plants grown in saline soil tended to overproduce the total soluble sugar and proline in plant tissues. The highest content of proline in two consecutive years (958.44 and 957.04 µg g$^{-1}$) was observed in PGPR-C3 consortia inoculated plants, followed by PGPR-C1 (799.07 and 800.03 µg g$^{-1}$), PGPR-C2 (690.04 and 699.52 µg g$^{-1}$), and uninoculated control plants (105.74 and 106.91 µg g$^{-1}$), respectively (Figure 7a). The average of both years' proline content was PGPR-C3 (957.74 µg g$^{-1}$) > PGPR-C1 (799.55 µg g$^{-1}$) >PGPR-C2 (694.78 µg g$^{-1}$) > Control (106.32 µg g$^{-1}$). Similarly, the highest content of total soluble sugar in both years (8.07% and 8.03%) was recorded in PGPR-C3-treated plants, followed by PGPR-C2 (5.47% and 5.43%) and PGPR-C1 (5.47% and 5.40%), significantly ($p \leq 0.05$) higher than uninoculated control plants (3.00% and 3.07%) according to Duncan's multiple range test.

**(a)**

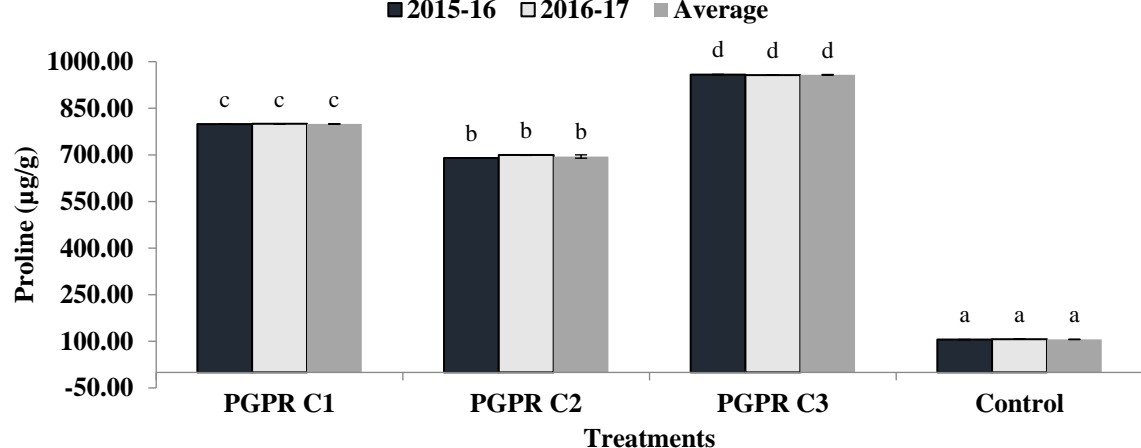

**(b)**

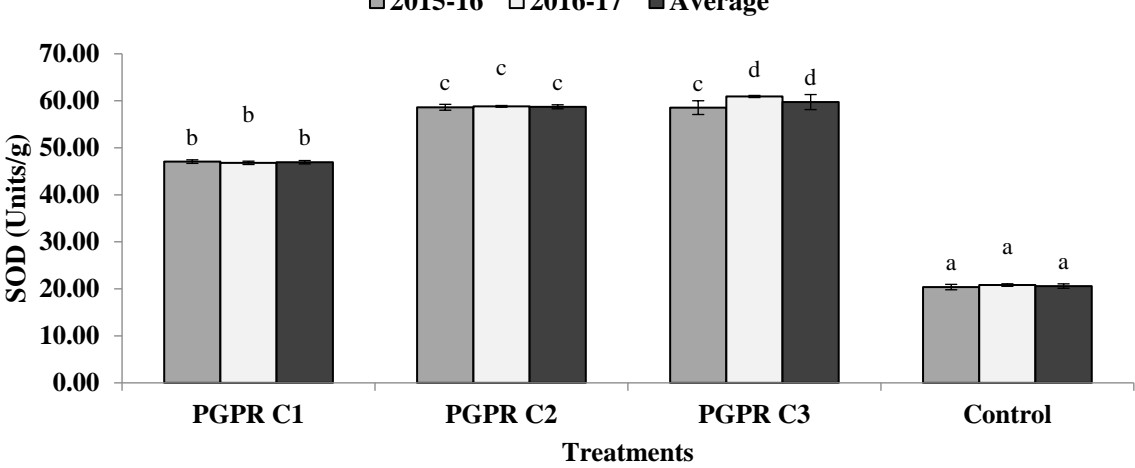

**(c)**

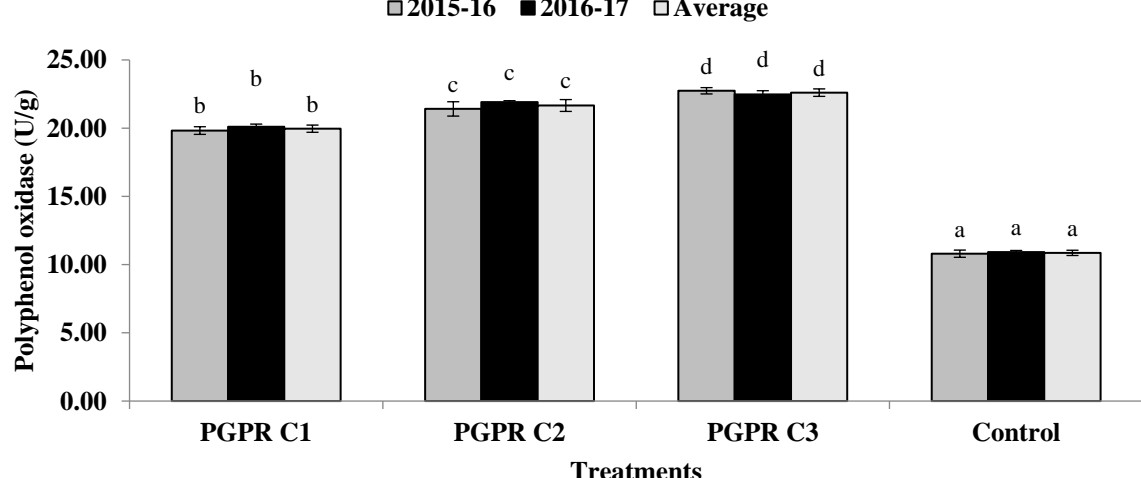

**Figure 7.** *Cont.*

**(d)**

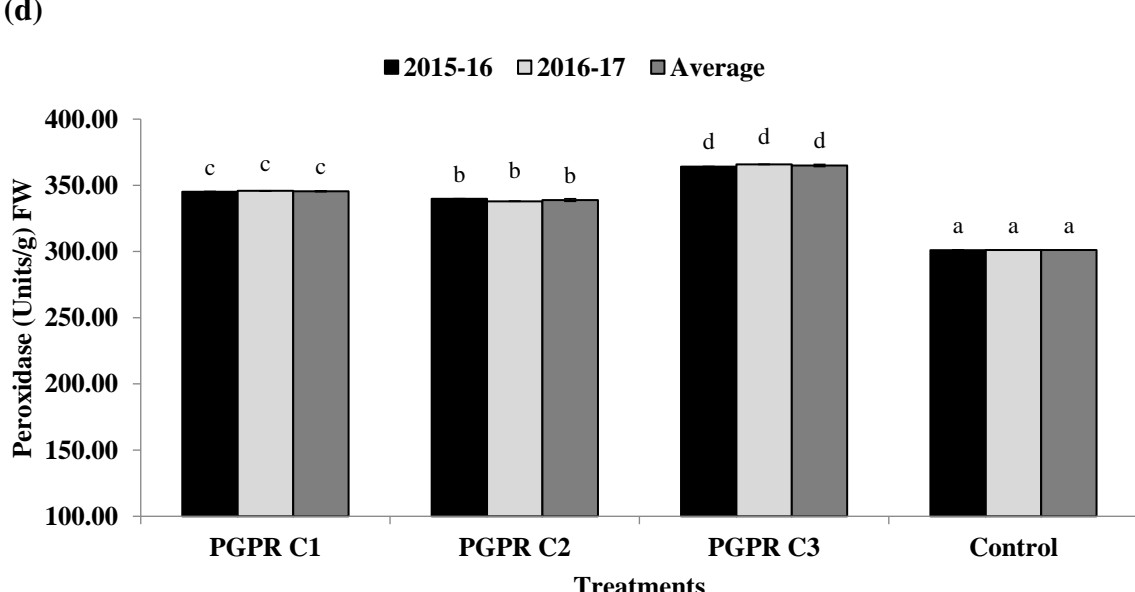

**Figure 7.** Effects of consortia of halotolerant PGPR on (**a**) Proline ($\mu$g g$^{-1}$ dry wt.), (**b**) Superoxide Dismutase (SOD) (Units g$^{-1}$ fresh wt.), (**c**) Polyphenol Oxidase (PPO) (Unit g$^{-1}$ fresh wt.) and (**d**) Peroxidase (POD) (Unit g$^{-1}$ fresh wt.) of tomato hybrid NS585 grown in the saline-stressed field of Central Soil Salinity Research Institute, Regional Research Station, Lucknow, Uttar Pradesh. PGPR-C1: (*L. fusiformis* (CSR-A11) + *B. subtilis* (CSR-A18) + *B. licheniformis* (CSR-M16)); PGPR-C2: (*L. fusiformis* (CSR-A11) + *L. sphaericus* (CSR-A16) + *B. subtilis* (CSR-A18)); PGPR-C3: (*L. fusiformis* (CSR-A11) + *L. sphaericus* (CSR-A16) + *B. licheniformis* (CSR-M16)). Values are mean of three replicates of two consecutive years (2015–16 and 2016–17) and average of two consecutive years (2015–2017) with $\pm$ standard deviation. Significant differences between treatments are shown by means denoted by different letters ($p \leq 0.05$) (DMRT).

The activities of the antioxidant enzymes (SOD, POD, and PPO) of all the microbial consortia inoculated plants invariably showed significantly ($p \leq 0.05$) higher antioxidant enzyme activity than the untreated control plants under salt-stressed conditions in sodic soils (Figures 6d and 7b). PGPR-C3 consortia inoculation significantly ($p \leq 0.05$) enhanced the SOD, POD, and PPO levels in the first year (58.55 U/g, 364.18 U/g, and 22.74 U/g) and second year (60.92 U/g, 365.81 U/g, and 22.47 U/g) compared to untreated control plants. Similarly, SOD, POD, and PPO levels (58.63 U/g, 339.87 U/g, and 21.41 U/g) in PGPR-C2-inoculated plants in the first year of the field trial, followed by (58.82 U/g, 337.93 U/g, and 21.91 U/g) in the second year, were significantly ($p \leq 0.05$) higher according to DMRT analysis compared to the control plants, but less than the PGPR-C3 consortia (Figures 6d and 7b).

## 4. Discussion

Salt stress is a strong production constraint, causing 20–50% yield reductions in important agri-horticultural crops [47]. Tomato is an important crop known to be sensitive to salt stress when soil pH is above 8.0 [48,49]. Therefore, there is a need to adopt innovative research strategies in addition to the existing organic or inorganic soil amendments to cultivate crops [14] successfully. Harnessing the rhizosphere diversity of the sodic soils for halotolerant rhizobacteria is an alternative management approach that has shown beneficial effects in alleviating the salt stress in the rhizosphere of crops like gladiolus, tomato, and wheat [50]. This study reports the assessment of the efficacy of potential H-PGPR isolates obtained from the sodic rhizosphere for the growth of tomatoes in pot and field experiments at soil pH of 9.23 and 9.14, respectively. Many workers previously reported the isolation of halotolerant PGPR from rhizospheric saline wheat soil [51,52]. Halotolerant rhizobacteria are associated with many plant growth and biocontrol abilities

that aid in the amelioration of soil salinity and induce tolerance against biotic stress [53]. *Lysinibacillus* spp. and *Bacillus* spp. used in this study, in addition to having salt tolerance properties, also produced extracellular enzymes, i.e., amylase, protease, cellulase, and lipase, that helped in the implication of the antifungal activities against the pathogenic microorganisms. These findings were in line with Karthika et al. [54], who showed the production of proteases, cellulases, lipases, xylanases, and β-1,3-glucanases by *Bacillus* sp. ruptures the cell wall of soil-borne pathogenic microorganisms.

Due to severe salt stress, higher levels of salt accumulation were observed in the cytoplasmic region of plant cells, which drastically affects enzymatic activity inside the cell and leads to metabolic changes resulting in an elevation of ethylene and abscisic acid concentration. In contrast, cytokinin and auxin hormone concentrations decreased [55]. Halotolerant PGPR tends to alleviate salt stress. Bacterial species such as *Pseudomonas* sp., *Bacillus* sp., *Enterobacter* sp., *Klebsiella* sp., and *Agrobacterium* sp. have been reported by many scientists as bio-inoculants with efficient halotolerant abilities used in saline agriculture [56]. In this study, we investigate the efficacy of the nine halotolerant bacterial isolates for growth and biomass production in tomatoes under sodic soil of pH 9.23 under pot culture experimentation. *Lysinibacillus* spp. and *Bacillus* spp. seed treatment and soil application significantly enhanced tomato vegetative parameters and biomass compared to the other isolates studied. The result supports the earlier findings of Damodaran et al. [29], which reported a high vigour index of the wheat plants treated with *Lysnibacillus* spp. and *Bacillus licheniformis*. The salt-tolerant wheat variety Aas-11 inoculated with *Bacillus pumilus* exhibited higher growth, fresh and dry biomass than *Pseudomonas fluorescence* and *Exiguabacterium aurantiacum* [57].

The isolates of *Bacillus thuringiensis* (CSR-B3) and *B. pumilus* (CSR-B2) promoted the growth and biomass of salt-stressed rice and gladiolus plants [34]. Endogenous ethylene levels in plants inoculated with effective rhizobacterial isolates are due to aminocyclopropane activity, resulting in the delay of senescence due to salt stress [58]. *Pseudomonas stutzeri* and *Klebsiella pneumoniae* were able to produce nitrogenase enzymes, indole acetic acid, phosphate solubilization, ammonia, and siderophore, which triggered plant growth to mitigate salinity stress in rice. Various metabolites were produced by halo-tolerant microbes, which protect plants from oxidative stresses exerted by soil salinity [59]. Consortia performed better than individual applications for inducing salt tolerance in plants and promoting growth.

Formulating compatible microbes of different genera into consortia significantly improves the growth of plants under saline conditions [60]. Previous studies also showed enhanced salt tolerance upon inoculating microbial consortia instead of using a single isolate [61]. The current study deals with the comprehensive screening of halotolerant PGPR in consortia form in tomato plants grown in salinity-affected soils. Many scientists have shown that halotolerant rhizobacteria isolated from salt-affected soils can survive in a highly saline environment [62]. If they have various plant growth-promoting potentials like higher production of ACC deaminase and IAA, they can play a vital role in combating salinity stress for sustainable agriculture. It has been previously documented that bacteria that encourage plant growth play a role in the phytoremediation of high salt-containing soil. In the current study, efforts were made to form different consortia of highly efficient halotolerant rhizobacteria obtained from the pot experiments and assess their field tolerance against soil sodicity (pH 9.14). Bio-inoculation of bio-consortia (PGPR-C3), comprised of CSR-A11 (*L. fusiformis*), CSR-A16 (*L. sphaericus*), and CSR-M16 (*B. licheniformis*), relieved the tomato plants from salt stress and significantly enhanced plant growth, root and shoot biomass, yield, and lycopene content. Our findings were concomitant with the previous studies that proved that inoculating PGPR consortia in wheat [56] and tomato [60] alleviated salt stress. It might be because PGPR has the potential for ACC deaminase metabolism, solubilizing phosphorous [63] and potassium [64]. Inoculation of *Paenibacillus polymyxa* and *Bacillus megaterium*, along with *Rhizobium*, significantly augmented the biomass of *Phaseolus vulgaris* [65]. Similarly, PGPR consortia of *Paenibacillus, Lysinibacillus*, and *Bacillus* strains

increased the chlorophyll content, plant biomass, and nutrient uptake in *A. thaliana* [66]. Under saline conditions, tomatoes co-inoculated with AMF (*arbuscular mycorrhizal* fungi) and *Pseudomonas fluorescens* C7 produced a high mean fruit weight [67]. An increase in tomato yield and lycopene content by the application of an AMF-based formulation under saline conditions compared to the control was also reported [68]. In the present study, the increment observed in the yield of tomato under saline conditions may be due to the enhanced nutrient uptake (phosphorous availability) and higher IAA production mediated by the inoculation of the microbial consortia, which is consistent with the findings of Mahanta et al. [69].

The Na/K ratio was an essential parameter in determining salt tolerance in solanaceous vegetables [26]. An increase in salinity and alkalinity significantly increased the Na/K ratio in the shoot [70]. Our study observed a significantly lower Na/K ratio in the bio-consortia (PGPR-C3)-treated tomato plants than in control. The microbial consortia treatment significantly enhanced the selective absorption of K$^+$ ions instead of Na$^+$ ions, which is involved in the induction of tolerance in tomato plants against salt stress. The suppression of Na$^+$ ion uptake by the high uptake of K$^+$ may be due to the antagonism between the two cations. The results were consistent with the earlier findings of Strahsburger et al. [71]. The treatment of the bacterial strains of *Bacillus pumilus* in gladiolus increased the uptake of K$^+$ ions compared to Na$^+$ ions, attributing to a low Na/K ratio in gladiolus [34]. The high accumulation ability of the combined application of PGPR isolates tends to increase the shoot length, fresh weight, K$^+$, and Ca$^{2+}$ content, where the synergistic role of microbial consortia is evident to alleviate the salt stress and increase the plant growth and yield in wheat [56], as we observed in our case with the tomato. The synergetic function controlled by PGPR was highly apparent in the reduction of ionic toxicity following salt stress exposure, which resulted in enhanced growth and yield of wheat.

In addition to the Na/K ratio, biochemicals such as proline and TSS are known to impart tolerance to the host in gladiolus under salt stress conditions [34]. In our study, total soluble sugar and proline accumulation varied significantly higher in the PGPR-C3-inoculated plants compared to uninoculated plants grown in salt-stressed soil. The results confirmed the previous findings of Upadhyay et al. [20] that halotolerant PGPR *Arthrobacter* sp. and *Bacillus subtilis* (SU47) enhanced the dry biomass, proline, and total soluble sugar content in the wheat crop grown under saline conditions. Inoculation of PGPR isolates consortium modulated tomatoes' physical and chemical characteristics under saline-stressed conditions. The synthesis of proline and total soluble sugar varied significantly in inoculated plants compared to uninoculated plants grown in salt-stressed soil. The results confirm the earlier findings of Upadhyay et al. [20] that halotolerant PGPR *Arthrobacter* sp. and *Bacillus subtilis* SU47 increased the dry biomass, proline, and total soluble sugar content in the wheat crop grown under saline conditions.

Apart from bio-inoculants' role in nutrient uptake, an enhanced plant's natural defensive system due to the host plant's increased enzymatic activity is also essential [72]. Under stress conditions, plants are prone to produce antioxidants to impart tolerance to the stress; however, in susceptible conditions, the inherent production of the plant antioxidants is not sufficient to induce tolerance. Efficient halo-tolerant bio-inoculants tend to induce the defense mechanism of the inoculated host through production of more antioxidants like proline, PPO), (POD), and (SOD) [73]. In the current experiment, the PGPR-C3 consortia-inoculated tomato plants showed higher PPO, POD, and SOD production than other microbial consortia. On the other hand, the uninoculated control tomato plants showed significantly low antioxidant production and plant growth parameters, confirming the role of the antioxidants as a potential defense mechanism adopted by bio-consortia-inoculated plants. The reduced antioxidant content is a prominent feature of yield and growth decline in salt-sensitive plants under salt-stress conditions [74].

Furthermore, root antioxidant enzymes negatively affect soil sodicity [75]. PGPR isolates with high antioxidant enzyme production abilities stimulate plants' antioxidative defense mechanisms, which remove free radicals produced by saline soil [76]. Thus, it

is evident that antioxidants play a vital role in the induction of tolerance for salt stress in plants.

The current study shows that inoculation of halotolerant PGPR enhanced the yield of tomato crops cultivated under salt-stressed conditions. These findings are in line with the previous studies, which demonstrated increased plant growth and production after treatment with halotolerant PGPR under salt stress in crops like wheat [77]. Thus, the formulation of a halotolerant microbial using the PGPR-C3 consortia is a promising tool to minimize the yield loss incurred due to salt stress.

## 5. Conclusions

Based on our research carried out for identification of the potential salt-tolerant PGPR isolate to enhance growth and yield of tomato in the sodic soils of pH > 9.0, application of the microbial consortia PGPR-C3-based formulation was influential in the enhancement of tomato fruit yield quality in the sodic soils fields. In light of the results obtained in the present study, it was evident that the enhanced growth promotion under salt-stressed conditions of sodic soils by the PGPR-C3 microbial consortia was attributed to its efficiency in selective absorption of potassium $K^+$ compared to $Na^+$ and ability to produce high antioxidant enzymes like POD, PPO, and biochemicals like proline. The results were further validated in the climate-smart sodic villages in the Karnal region of northern India. Thus, the current study dealt with developing effective halotolerant microbial consortia PGPR-C3 for enabling sustainable cultivation of essential vegetable crops like tomato in sodic soil.

**Author Contributions:** Conceptualization, T.D.; Methodology, T.D., S.K.J. and S.K.; Validation, T.D. and A.S.; Formal analysis, H.S.J.; Investigation, G.G. and R.G.; Resources, V.K.M. and P.C.S.; Data curation, T.D. and S.K.; Writing—original draft, T.D. All authors have read and agreed to the published version of the manuscript.

**Funding:** This research work was funded by Indian Council of Agricultural Research (ICAR), Govt. of India, New Delhi under ICAR—AMAAS Network Project Scheme sanctioned to T.D. Author G.G. is thankful to ICAR, New Delhi for providing research fellowship (Senior research fellowship).

**Institutional Review Board Statement:** Not Applicable.

**Informed Consent Statement:** Not Applicable.

**Data Availability Statement:** Data is contained within the article.

**Acknowledgments:** The authors thank the ICAR—AMAAS Network Project, Mau, for providing the research opportunity, lab support, and other necessary facilities.

**Conflicts of Interest:** The authors declare no conflict of interest.

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
