# Peer review of "Development of Halotolerant Microbial Consortia for Salt Stress Mitigation and Sustainable Tomato Production in Sodic Soils: An Enzyme Mechanism Approach"

_sustainability, doi:10.3390/su15065186_

Round 1

Reviewer 1 Report

The authors collected 9 bacterial strains, and investigate their promoting ability on plant health and growth. Furthermore, by conducted biological and biochemical assays, the study suggested PGPR-C3 which contains CSR-A11, CSR-A16 and CSR-M16) has promoting ability on tomato growth, salt tolerance and the accumulation of Lycopene. This provide a valuable knowledge to the management of crop farming. However, there’s some detailed information that is missing, and it caused confusion while reading. Please see below:

(1)   Please provide the information of isolates used in this study, which may include previous published articles if there’s any or detailed isolation method.

(2)   I recommend Table 1 and Figure 1 can combined into a figure (Bar figure). The current version seems unnecessarily duplicated.

(3)   Please add the significance test to Figure 3 if you want to mention significant decrease in main text.

(4)   I suggest Table 4 may be converted in Bar figure

(5)   Please fill the correct text in “The institutional review board statement”, “informed consent statement” and “data availability”.

Author Response

Thanks for the good suggestions. The comments have been addressed as per the suggestions. 

Reviewer 2 Report

This review is concerning a research work entitled “Development of halotolerant microbial consortia for Salt Stress mitigation and Sustainable tomato production in Sodic soils:  An enzyme mechanism approach” By Damodaran, et al. As detailed data and interpretation, I recommend it for an international audience in this journal, however several points have to be precised and a major revision is requested.

The major points are:

1.    I put in the minor points at the end some of these, but the whole text has to be considered.

2.    Make sure that all scientific names in the References list are italics.

3.    All tables must be self-explanatory.

4.    You do not need to use abbreviation in the abstract

5.    Abstract should mentioned some results

6.    In many places in the MS you used word with capital letter

7.    Line 89-91 please check it and rewrite again

8.    Line 83 please check it

9.    Line 101 please write the method in brief

10.    Line 104 what the meaning autoclaved and sterilized

11.    Line 103 from where you got the seeds?

12.    Line 108- 110 as you mentioned before this isolated were identified by authors 29 are you re identified it again

13.    Line 217-228 this part in in M&M

14.    References need to be revise to fit with the Journal style

15.    I added some more comment in attach file please see it and revise according the comments

Author Response

(The authors gave the same response as above.)

Reviewer 3 Report

Why author had used old data to represent the outcome..

There is no pool data analysis of two year.

Why author used 9 PGPR ? what is the significance and characteristics features of 9 isolates ?

Why author needs to add 9 isolates and make consortia ? sometimes lower number with diverse charactertistics alsl works.

The medium formulation of consortia is always an issue as there is growth competition in the consortia.

Abstract line 19. How does enzymes enhances yiled and growth as all microbes produces hydrolyzing enzymes.

Why author did only two PGPR test  is there any significance of only two test?

If they are PGPR with different beneficial traits then why author studied their enzymatic profiles? Moreover as per results shown in table 2, surprisingly  most of them does not have these secretory protein profle.

Bacillus and pseudomonas which are industrial sources of enzymes did not shows activity …It seems to be surprisingly.

2.2 method chapter …citation is not proper as it leads to confusion for author “ The appearance of clear zones around  bacterial growth after overlaying culture plates with Gram's iodine was considered to be positive for cellulase activity. Similarly, proteolytic activity was analyzed as per [31] with” like wise in many parts of MS

Method 2.3 how large amount of soil is sterilized ?

Describe CSR media

What author mean by writing 1%  “he prepared bacterial suspension was used for treating the tomato seeds with 1% for 2 h and further, for foliar application at 20th day after sowing.

Method section2.3..Author has already organism with acesion number available than why they wants to go for BLAST and 16s rRNA analysis.

Even author had done this than also the part of molecular analysis is missing?

Normallly tomato plants are transplanted and this information is missing.

2.3 how constant stress was exerted and why for 40 days . is their any reports ?

2.4 should be come first and then 2.3 section.

Name of instrument , make and year is missing throughout the MS

Sction 2.6 spectrophotometer at 600 nm (OD600 …why repeated

supplemented with 5.0 % (0.5 g), 7.5 % (0.75 g) and 10.0 % (1.0 g) NaCl….5 gram itself define 5% ..no need to write again

why 2.6 and 2.7 as it has been reported to be PGPR mentioned initially in MS

2.8 section is just repeattaion..

Section 2.10 is not properly written and citations is not prper.

Results …when accretion number is assigned and given in method section and do not repeat again…use code given by the author

Selexction of consortia based on enzymatic profile is very weired concepts as they might have ceratin features that enhances the growth of plants…

Table 3 ..when dmrt is given then no need to write Sem in bracket ..very wrong way of presenetation

Figure 1 … no statistical analysis

Same with figure 2

Author Response

(The authors gave the same response as above.)

Round 2

Reviewer 2 Report

authors has been covered all my previously comments 

good luck 

Author Response

Dear Dr.

Thank you very much for your excellent remarks and guidance in redefining the manuscript with a new vision. 

Reviewer 3 Report

I don't find any improvement in the paper as the author tried to convince others what they believe. I mentioned why did the author take 6 years to communicate the paper. There are unanswered questions raegarding enzymes production and only two tests of PGPR. bacillus and pseudomonas can be pathogenic?? HOw author assure that they are nonpathogenic ...

There are lot of unanswered queries..

Author Response

Dear sir,

Thanks for your valuable comments. We have tried our best to address the issues and request you to kindly accept the responses provided to the comments
